# Prospective Screening of Cancer Syndromes in Patients with Mesenchymal Tumors

**DOI:** 10.3390/cancers16223816

**Published:** 2024-11-13

**Authors:** Ingegerd Öfverholm, Yingbo Lin, Julia Mondini, John Hardingz, Robert Bränström, Panagiotis Tsagkozis, Valtteri Wirta, Anna Gellerbring, Johan Lindberg, Venkatesh Chellappa, Markus Mayrhofer, Cecilia Haglund, Felix Haglund de Flon, Karin Wallander

**Affiliations:** 1Department of Oncology-Pathology, Karolinska Institutet, 17164 Stockholm, Sweden; 2Department of Clinical Genetics, Karolinska University Hospital, 17176 Stockholm, Sweden; 3Biotechnology Unit, Linköping University, 58183 Linköping, Sweden; 4Department of Molecular Medicine and Surgery, Karolinska Institutet, 17176 Stockholm, Sweden; 5Department of Breast Cancer, Endocrine Tumors and Sarcoma, Karolinska University Hospital, 17176 Stockholm, Sweden; 6Department of Endocrine and Sarcoma Surgery, Karolinska University Hospital, 17176 Stockholm, Sweden; 7Science for Life Laboratory, Department of Microbiology, Tumor and Cell Biology, Karolinska Institutet, 17165 Stockholm, Sweden; 8Science for Life Laboratory, School of Chemistry, Biotechnology and Health, Royal Institute of Technology, 10044 Stockholm, Sweden; 9Genomic Medicine Center Karolinska, Karolinska University Hospital, 17164 Stockholm, Sweden; 10Department of Medical Epidemiology and Biostatistics, Karolinska Institutet, Nobels väg 12 A, 17177 Stockholm, Sweden; 11National Bioinformatics Infrastructure Sweden, Science for Life Laboratory, Department of Cell and Molecular Biology, Uppsala University, Husargatan 3, 75237 Uppsala, Sweden; 12Department of Pathology and Cancer Diagnostics, Karolinska University Hospital, 17176 Stockholm, Sweden

**Keywords:** sarcoma, germline, second hit, ATM, KCNQ1, CDC73, MLH1, MSH6, POLG

## Abstract

Tumors in soft tissue and bones are rare, and there is limited knowledge about how they occur. Better knowledge about inherited predisposition to tumor syndromes increases the chance for the medical community to detect cancer early through targeted screening programs and to choose the most appropriate cancer treatment. In our study, we show that some inherited mutations can increase the risk for these tumors. We applied a novel approach, in which we analyzed a blood sample in tandem with a tumor sample from participants, and we found especially interesting inherited mutations. Some of the mutations we found were already known to increase the risk of cancer, although not proven to be connected to soft tissue and bone tumors before. Other mutations we present have not been shown to be connected to tumors at all before. Our findings can guide further genetic investigations of soft tissue and bone tumors.

## 1. Introduction

Mesenchymal tumors such as lipomas, uterine leiomyomas, and fibrous histiocytomas are common in the general population. Over a hundred subtypes have been described, many having distinct genomic alterations such as fusion genes, small nucleotide variants, or copy number alterations [1]. Malignant mesenchymal tumors (sarcomas), on the other hand, are extremely rare and constitute less than 1% of all malignancies. Compared to carcinomas, sarcomas more commonly affect children and adolescents [2]. Their etiology is generally unknown, and most sarcomas are considered sporadic, without association to germline variants [3]. However, tumor syndromes such as neurofibromatosis type 1 may increase the risk of certain types of mesenchymal tumors [4], and some cancer syndromes have an established increased risk of sarcomas. For example, heritable *TP53*-related cancer syndrome is associated with an increased risk of osteosarcomas and soft tissue sarcomas [5], and heritable retinoblastoma syndrome (with germline pathogenic variants in the *RB1* gene) increases the risk of osteosarcomas, rhabdomyosarcomas, and leiomyosarcomas [6].

We have previously described the impact of whole genome sequencing (WGS) and whole transcriptome sequencing (WTS) on sarcoma diagnostic classification and treatment prognostic biomarkers in clinical practice [7]. Such analyses can potentially reveal pathogenic germline variants, which may be associated with tumor development. Here, we describe an expanded patient cohort from a germline perspective, in which both germline WGS, tumor tissue WGS, WTS, and methylation analyses are interpreted in the light of patient phenotype (see Appendix A for the study pipeline).

## 2. Materials and Methods

### 2.1. Study Population

This prospective clinical research study offered inclusion for any patient treated at the Karolinska University Hospital in Stockholm, Sweden, with a primary or metastatic tumor suspicious for sarcoma during the years 2022 and 2023. The predominance of adult patients was expected in advance, since childhood sarcomas were mainly analyzed in another pipeline as part of an ongoing parallel study [8]. Family history or history of other tumors did not influence study inclusion. The research was approved by the Swedish Ethical Review Authority (ID 2022-05409-01 and 2013 1979-31 with amendment 2018/2124-32) and was designed in accordance with Swedish law and the Declaration of Helsinki. All participants, or their legal guardians for minors, had given their informed consent prior to their enrollment in the study.

### 2.2. Whole Genome and Transcriptome Analysis

DNA and RNA isolation from blood and tumor tissue, library generation, and sequencing were performed as previously described [7]. In short, DNA and RNA libraries were generated with the Illumina TruSeq DNA PCR-Free library preparation kit, and the Illumina Stranded mRNA Prep (Illumina, San Diego, CA, USA), respectively. Sequencing was performed on the Illumina NovaSeq 6000 or NovaSeq X Plus platforms using paired-end 150-bp sequencing, with at least 30x coverage in normal DNA and 90x in tumor DNA.

Germline WGS data were analyzed with the pipeline Mutation Identification Pipeline (versions ranging from 11.0.2 to 12.0.3) [9,10], and somatic WGTS data were analyzed with the BALSAMIC (version 12.0.2) [11] and AutoSeq pipelines [12], as previously reported [7].

### 2.3. Identification and Reporting of Germline Variants

Germline variants were limited by in silico filtering to 787 genes with known or suspected association to hereditary cancer (Appendix A). The gene panel was the result of a curated list of genes with potential links to cancer syndromes previously published by our group [13] and the human phenotype ontology term “neoplasia” (HP:0002664) accessed January 2022 [14]. Manual filtering of all coding (exonic or splicing) germline variants with a gnomAD [15] allele population frequency of <0.01 was performed. Variants were excluded if A) they were reported as likely benign or benign in ClinVar [16] by more than two submitters, or B) they occurred in the local variant database (9244 cases) > 100 times and were synonymous, or C) they were reported as likely benign or benign in ClinVar by only one submitter but occurred in the local variant database > 80 times.

After initial variant filtering, clinical assessment was manually performed based on participant phenotype and the American College of Medical Genetics and Genomics and the Association for Molecular Pathology (ACMG/AMP) criteria [17]. Missense variants with no reported association to a cancer syndrome in ClinVar [16] and potentially truncating variants (nonsense, frameshift, or splice site) with either no association to the phenotype of the patient, or in genes without truncation being the known mechanism of disease, were considered variants of unknown significance. All patients with a pathogenic or likely pathogenic variant were further checked for medical history, known cancer or other traits in the family, and previous medical treatment or specific carcinogenic exposure. Pathogenic or likely pathogenic variants (heterozygous for dominant disorders and homozygous or compound heterozygous for autosomal recessive disorders) determined to be clinically actionable were reported to the referring physician, who informed the patient and offered further referral for genetic counselling. Germline variants associated with a phenotype known or previously suggested to increase the risk of mesenchymal tumors were labeled “second hit expected”, and the remaining genes without a known association were labeled “second hit not expected”.

### 2.4. Identification of Somatic Variants

Genes with pathogenic or likely pathogenic germline variants were analyzed for somatic variants in the tumor DNA, including small nucleotide variants (SNVs), copy number alterations (CNAs), loss of heterozygosity (LOH), and inactivation by structural variants. Filtering included a variant allele frequency of >5% for SNVs/deletions and >10% for structural variants. All variants were manually inspected in the integrated genome viewer (IGV) [18]. Somatic CNAs and LOH were visually addressed in AutoSeq [12,19]. Chromothripsis-like CNA profiles and large deletions were considered uninformative. Tumor mutational burden (TMB) was calculated from all somatic variants that had passed quality control (i.e., minimal read depth 10, minimal allele frequency 0.05, maximal gnomAD frequency 0.1%) and divided by 3000 Mb. TMB > 10 variants/Mb was considered “TMB-high”. Germline variants with no previously known association to the phenotype but with detectable same-gene second hits were considered novel, and these genes were further checked in the Catalogue Of Somatic Mutations In Cancer (COSMIC) database and The Cancer Genome Atlas (TCGA) datasets for further assessment of potential impact in soft tissue tumors [20,21]. For all patients with germline variants without a confirmed second hit, we expanded the search of second hits to include genes whose proteins were associated with the germline gene (including the affected signaling pathway and proteins with known interactions with the gene products).

WTS alignments (.cram files) from the RNA fusion pipeline were assembled into potential transcripts with StringTie v2.2.0. Gene expression values (FPKM) for the genes of interest were extracted. Aberrant RNA expression was determined by plotting the gene expression of the whole cohort (312 cases). Extremely low or high expression (top or bottom 5th percentile) for a gene with a germline pathogenic hit was considered aberrant and indicative of a second hit.

### 2.5. Methylation

Genomic DNA from fresh frozen tumors was treated with sodium bisulfite using the EZ-96 DNA methylation kit (Zymo Research, catalogue number D5004), following the manufacturer’s standard protocol. An assessment of the levels of DNA methylation of known CpG regions and promoters across the genome was performed with the Infinium MethylationEPIC v2.0 Kit (Illumina, San Diego, CA, USA) and Illumina iScan. In brief, following bisulfite conversion, approximately 500 ng of the bisulfite-converted DNA per sample was used for methylation analysis. The initial quality control and identification of signal intensities for each probe were performed with Illumina GenomeStudio Software 2011.1.

### 2.6. Statistical Analyses

Statistical analyses were performed in Python with the package SciPy. Student’s *t*-test was used for the comparison of age distribution. Odds ratios were calculated for malignancy versus non-malignancy and woman versus man. A *p*-value < 0.05 was considered significant.

## 3. Results

### 3.1. A Minority of Sarcoma Patients Have Tumor Predisposition Syndromes

In total, 316 patients with a preoperative suspicion of sarcoma were prospectively included in the study. After final histopathological analysis, four cases were excluded from the study (two patients with malignant melanoma, one with endometrial carcinoma, and one with colorectal cancer). This generated a study cohort of 209 (67%) sarcomas, 40 (13%) gastrointestinal stromal tumors (GISTs), and 63 (20%) benign mesenchymal tumors. A summary of all final diagnoses is available in Appendix A.

We performed whole genome sequencing from a peripheral blood sample and multiomics including DNA, RNA, and methylation analyses of the corresponding tumor tissue sample. In 8% of the whole cohort (24/312), a pathogenic or likely pathogenic germline variant associated with a dominant hereditary tumor syndrome was detected (Table 1). When excluding benign cases, the tumor syndrome detection rate was similar (8%, 21/249). Most of the carriers had no medical history of previous cancer, and no known cancer in the family. A minority of carriers would have fulfilled the testing criteria for their specific cancer syndrome (42%,10/24), and two of them had regardless not been offered a germline genetic screening. No homozygous or compound heterozygous variants in genes associated with a recessive syndrome were detected.

Patients with pathogenic or likely pathogenic germline variants had somewhat different tumor diagnoses than the rest of the cohort (Table 2), with a greater ratio of leiomyosarcomas. There were no significant differences regarding age distribution, malignant tumors, or sex ratio between the group with germline pathogenic findings and the group without (Appendix A).

### 3.2. Many Germline Variants Currently Have Limited Clinical Utility

Of the 24 patients with a potential hereditary tumor syndrome, 16 had findings that were considered clinically actionable, leading to genetic counselling and carrier testing in the family. Half of this group received surveillance for mesenchymal tumors and the other half for carcinomas only.

The remaining eight pathogenic and potentially actionable germline variants were not reported to the treating clinicians, in accordance with ACMG/AMP criteria and current Swedish National Guidelines. These eight variants were either associated with potential hereditary tumor syndromes without any surveillance recommendations based on patient phenotype and pedigree (variants in the genes *CHEK2*, *EXT1*, *ATM*), or they were weakly associated with a condition potentially increasing the risk of cancer without being likely causative for a cancer syndrome in this specific case (variants in the genes *MRE11*, *BRIP1*, *ATR*, *DDX41*, *MITF*, *SDHAF2*).

### 3.3. Somatic Analysis Confirms Biallelic Inactivation and Establishes Sarcoma Syndromes

For patients with a pathogenic germline variant, we used the multiomics results from the corresponding tumor tissue sample, including whole genome and transcriptome sequencing and methylation analysis, to search for a second hit in the tumor. The initial search focused on the gene affected by a first, potentially causative germline hit. A pathogenic second hit was detected in the tumor from 11 of these patients (11/24, 46%), as presented in Table 1. In addition to this, two patients with germline *MSH6* pathogenic variants without detectable second hits had indirect signs of deficient MSH6: one had a mismatch repair (MMR) deficient tumor, and the other (who had received neoadjuvant treatment with complete response before the tumor sampling) had a preoperative biopsy showing complete loss of MSH6 immunoreactivity, consistent with deficient MMR.

All second hits were either missense or nonsense SNVs or focal deletions. No methylation aberration was detected as a second hit. Two illustrative examples of germline tumor suppressor inactivation with a somatic second hit and corresponding gene expression, as detected by multiomics in diagnostic patients, are depicted in Figure 1 and Figure 2.

Genetic counselling was recommended for all patients whose tumors harbored a second hit, except for the *ATM*, *POLG*, and *KCNQ1* carriers. Most of the germline variants with a second hit had previously been linked with an increased risk of mesenchymal tumors, or an association had previously been suggested, and were thereby considered as “second hit expected” (Figure 3). Vice versa, all patients with a germline hit considered “second hit not expected” (in genes without a previously known association with mesenchymal tumors or sarcoma), except for *CDC73* and *ATM*, had no second hit (Figure 3). For Lynch syndrome, the second hit was considered semi-expected based on the current literature.

For one female patient with a history of paraganglioma, pulmonary hamartomas, and SDH-deficient GIST (as determined by SDHB loss in the immunohistochemistry analysis), and no detectable germline variant, somatic methylation analysis detected biallelic hypermethylation of the *SDHC* promoter. This is consistent with the constitutional hypermethylation of *SDHC* that results in the somatic syndrome Carney triad (Figure 4).

The common c.1100delC (current nomenclature c.1229del) variant in the *CHEK2* gene was detected in three participants, and none of them had a second hit in *CHEK2*. Two of them, a woman with a history of breast cancer and a woman whose sister had breast cancer, were referred for genetic counselling, whereas the third case was a man without a first-degree relative with breast cancer, thereby not fulfilling the criteria for surveillance recommendation according to Swedish national guidelines [22].

In cases with germline pathogenic findings but without detectable second hits, genes with functions in the same pathways as the germline gene were assessed. No additional pathogenic somatic lesions were detected through this approach.

We also performed second hit analysis for 12 patients with pathogenic or likely pathogenic germline variants without established association with hereditary tumor syndromes (Appendix A). Of these, second hits in the tumors were identified in two patients, harboring germline variants in the *KCNQ1* and *POLG* genes and diagnosed with fibromatosis and GIST, respectively. In patients with heterozygous germline variants in genes associated with a recessive disorder, no second hits were detected in the tumor.

## 4. Discussion

In this prospective cohort of 312 patients with mesenchymal tumors suspicious for sarcoma, we confirmed a causal relationship between the tumors and an underlying tumor syndrome in 4% of the patients. An additional 3% of patients had germline variants associated with tumor syndromes or an increased risk of tumors, but without second hits, suggesting these variants were unrelated to the current diagnosis.

The prevalence of tumor predisposition in our cohort is similar to previous publications, with differences related to the clinical assessment of actionability [23,24].

Studies of potential causative mesenchymal tumor syndromes are sparse. Recently, Ballinger et al. published the largest study so far regarding sarcoma genetic predisposition. They identified two sarcoma-specific pathways involved in mitotic and telomere functions and concluded that further studies are needed to map their connection to an increased risk of sarcomas [25]. In our study, 16 individuals were referred for genetic counselling, and carrier testing was offered to their family members. Even though the risk of sarcomas is usually low for most cancer syndromes, thereby not warranting surveillance programs, the concurrent risk of developing a carcinoma might be high. Additionally, some germline findings directly impact the sarcoma treatment, such as avoidance of radiation therapy in the case of pathogenic *TP53* variants and potential immunotherapy for Lynch syndrome patients [26,27].

To differentiate causative germline variants from incidental findings in a diagnostic setting, we applied a somatic second hit omics approach. In clinical practice, this procedure is not widely used, as it requires both a tissue biopsy and a blood sample. However, with the growing use of comprehensive genetic screening for tumors, using blood samples as a normal reference to filter for somatic driver events, the potential for this approach is expanding.

Several groups have shown the usefulness of evaluating germline findings in the light of second hits, showing that the rate of detectable same-gene second hits was higher in patients carrying a germline variant with a known association to their tumor than those with a non-associated tumor [24,28,29,30]. Yap et al. published a retrospective study with the aim to evaluate whether the identification of germline variants adds benefit to paired tumor/normal sequencing. They found that 7.3% of their large cohort harbored pathogenic or likely pathogenic variants, but for tumor types lacking genetic testing guidelines, such as sarcomas, this proportion was lower [24]. Also, second hits were only detected in a minority of the cases, for instance only in 29% of the *NF1* carriers and 31% of the *MSH6* carriers. Fiala et al. found that 12% (28/229) of pediatric patients with sarcoma had a germline pathogenic or likely pathogenic variant, including both those associated with dominant and recessive inheritance. In 81% of the cases with an expected germline finding, a second hit was found [29]. In a recent study by Tesi et al., the prevalence of childhood cancer predisposition in a cohort of children with tumors was 11% (35/309), with a second hit and/or a relevant mutational signature detected in 19/21 (90%) of tumors with informative data [8].

In this study, the second hit approach suggested *ATM* and *POLG* as new candidates for association with sarcoma and identified a potential association between *KCNQ1* and desmoid fibromatosis.

ATM is a tumor suppressor gene with multiple protein functions, such as DNA-repair and cell cycle regulation. In current clinical practice, this gene is classified as a moderate penetrance gene associated with an increased risk of breast cancer [31]. There is growing evidence that pathogenic *ATM* variants increase the risk of several tumor types, including melanoma, ovarian, and pancreatic cancer, among others [31,32,33]. In Sweden, only truncating *ATM* variants are considered actionable [22], but the missense variant detected in our study is a Finnish founder, with an increased risk of breast cancer [34]. The somatic second hit detected in our patient, c.5188C>T, is a well-established truncating pathogenic germline variant [16]. In the COSMIC database, somatic *ATM* variants are reported in 8% (82/1077) of soft tissue tumors, of which 16 are angiosarcomas. The c.5188C>T variant is reported in 13 cases (carcinomas (N = 9), malignant melanomas (N = 3), and one carcinoid) [20]. Aberrations in *ATM* are present in 5% (12/255) of the PanCancer Atlas Sarcoma dataset, mostly in myxofibrosarcomas [21]. The detection of the biallelic loss of ATM in an angiosarcoma tumor in our study contributes to the growing knowledge about ATM in tumorigenesis.

The *POLG* gene is primarily known to cause the autosomal recessive Mitochondrial DNA Depletion Syndrome 4B, but autosomal dominant inheritance of progressive ophtalmoplegia has also been described [35]. There is so far no known connection to sarcomas, but there are limited reports suggesting tumor suppressor properties [36]. The *POLG* germline variant identified in our study is reported as pathogenic in mitochondrial disease cohorts by multiple submitters in ClinVar [16], and somatic variants in *POLG* are reported in 1% (10/902) of soft tissue tumors in COMIC, of which 30% (3/10) are GISTs [20]. A second hit in a GIST is intriguing, and further studies are needed to determine the functional role of biallelic *POLG* inactivation in tumor development.

Pathogenic (mainly truncating) variants in the *KCNQ1* gene lead to autosomal dominant arrhythmia syndromes and recessive Jervell and Lange–Nielsen syndromes [37]. Both the germline and somatic variants detected in our patient are known pathogenic variants associated with long QT syndrome, and none of these specific variants are reported in tumor samples in the COSMIC database. However, other *KCNQ1* variants are reported in a variety of tumor types, including 4% (33/867) of soft tissue tumors, mainly in unclassified sarcoma from fibrous tissue/uncertain origin (23/33) [20]. Missense variants in the same gene have been associated with inherited gingival fibrosis [38,39]. Mice carrying a targeted deletion in the *KCNQ1* gene developed significantly more intestinal tumors than non-mutant mice, and low expression of KCNQ1 was associated with poor overall survival for colorectal cancer patients [40]. There are no reports of *KCNQ1* involvement in desmoid fibromatosis, and our patient’s tumor harbored the well-characterized *CTNNB1* somatic activating variant. There is evidence that KCNQ1 is involved in the regulation of the Wnt/b-catenin signaling pathway, which is normally upregulated and serves as the primary driver of desmoid fibromatosis [41]. This suggests a potential mechanistic link between the biallelic loss of *KCNQ1* and desmoid fibromatosis in this case. However, further studies are needed to clarify this connection.

These new candidates for mesenchymal tumor syndromes highlight the potential for rare, previously unrecognized low-penetrance syndromes.

The somatic second hit approach also verified pathogenic variants in the genes *SDHA*, *CDC73*, *NF1*, *RB1*, *TP53*, and *EXT2* as causative in their carriers. Germline variants in the *CDC73* gene, including the truncating variant detected in our patient, are known to cause hyperparathyroid-jaw tumor syndrome (HPT-JT), which has variable penetrance for parathyroid carcinomas, ossifying fibromas of the jaw, and uterine lesions (adenofibromas and rarely adenosarcomas) [42]. While the truncating somatic variant detected in our patient has been reported solely in parathyroid carcinoma (N = 4) in the COSMIC database, other somatic variants in *CDC73* are reported in 2% (30/1679) of soft tissue tumors [20]. Our findings further substantiate the association between germline *CDC73* variants and the infrequent development of adenosarcoma.

For *MLH1* and *MSH6*, our results support previous studies suggesting that sarcomas are rare manifestations of Lynch syndrome. Similar to the cases in our study, a small subset of sarcomas presenting in Lynch syndrome have been radiation-induced [43,44].

We considered the pathogenic variants in genes with no detected second hit to be unsolicited findings. For instance, the risk of mesenchymal tumors in *CHEK2* c.1100delC carriers is not known. Näslund-Kock et al. found a sex-adjusted hazard ratio for heterozygous carriers compared to non-carriers of 3.45 (95% confidence interval, 1.09 to 10.9) for sarcomas [45], which was not significant after correcting for multiple comparisons. Bychkovsky et al. showed that pathogenic variants in the *CHEK2* gene are not associated with an increased risk of sarcomas [46]. Abdelghani et al. found that 6/300 pediatric cancer patients had germline *CHEK2* pathogenic variants, one of whom had Ewing sarcoma, while none of the others had mesenchymal tumors [47]. Of course, a germline pathogenic variant might be associated with an increased risk for a tumor regardless of a somatic second hit. The second hit approach is merely one of the available tools for interpreting the pathogenicity of genetic variants. The ACMG/AMP criteria [17], including the statistical correlation between diagnosis frequency and carrier frequency, bioinformatic information such as how conserved the affected amino acid is, etc., are widely used in the clinic. For mesenchymal tumors, however, their rarity makes it harder to achieve statistical power and to prove a phenotype–genotype correlation.

The diverse nature of our cohort reflects the typical presentation of patients at a sarcoma reference center, including benign tumors suspicious for sarcoma based on radiological or clinical findings and true sarcomas. When we analyzed different subgroups separately, leiomyosarcomas and GIST were the most prevalent within the cohort with germline pathogenic findings, which is to be expected since these are also among the more common diagnoses. Two participants with leiomyosarcoma carried pathogenic variants and same-gene second hits in *RB1* and *TP53*, which have known connections to this condition [48,49,50], and one carried a pathogenic variant without a detectable in-gene second hit in the *DDX41* gene, without any known connection to leiomyosarcoma. Among the patients with GIST, four had germline pathogenic findings. In total, there were five participants with GISTs without mutations in *KIT*, *PDGFRA*, or *BRAF* (sometimes referred to as “wild-type GISTs”), of whom two had germline pathogenic findings (in the genes *SDHA* and *NF1*). The number of wild-type GIST patients in our study is too small to base any statistical analyses on. Mandelker et al. reported a cohort with 35 wild-type GIST patients, in which they found germline pathogenic variants (in the genes *SDHA*, *SDHB*, *SDHC*, *NF1*, and *KIT*) in 70% (24/35) [51]. These results highlight the importance of germline genomic analysis for this patient group.

## 5. Conclusions

In this prospective study, we screened patients at a sarcoma reference center for germline variants. A significant proportion of patients (4%) harbored germline variants associated with a tumor predisposition syndrome and a second hit in the tumor. Pathogenic germline variants and somatic second hits were found in *NF1*, *RB1*, *TP53*, *EXT2*, and *SDHC*, in patients with a malignant peripheral nerve sheath tumor and GIST, leiomyosarcoma, leiomyosarcoma, secondary peripheral chondrosarcoma, and GIST, respectively. Both germline and somatic hits were found in the *ATM*, *CDC73*, *MLH1*, *MSH6*, *POLG*, and *KCNQ1* genes, which have no previously known connection with sarcoma. As WGS becomes routine clinical practice for sarcomas and other rare tumors, an integrated somatic–germline analysis is a feasible and efficient approach to evaluate germline findings in the clinical setting, and it could be employed in the reality of the rapidly expanding WGS analyses performed in cancer centers. Importantly, germline variants need to be interpreted in their clinical context, and knowledge about how to handle unsolicited findings and variants of unknown significance is crucial.

## Figures and Tables

**Figure 1 cancers-16-03816-f001:**
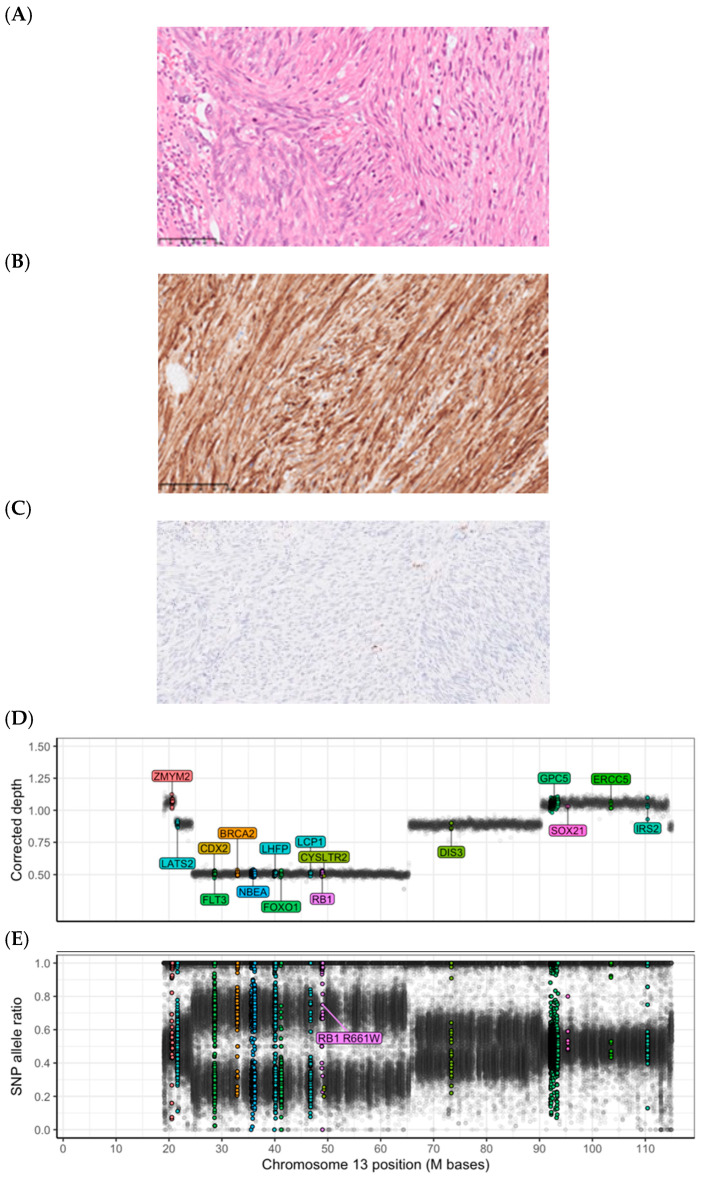
Histopathology and genetics of an *RB1* germline-related leiomyosarcoma. Patient 101 carried a germline *RB1* pathogenic variant causing hereditary retinoblastoma syndrome. The tumor harbored a deletion of the *RB1* locus (13q), resulting in the loss of heterozygosity and biallelic inactivation. (**A**–**C**): Microphotographs show (**A**) a routine hematoxylin–eosin stain of a leiomyomatous tumor with high grade atypia, and immunohistochemistry showing (**B**) positivity for desmin, and (**C**) a loss of Rb immunoreactivity (single cells with retained Rb expression are tumor-infiltrating immune cells), as discovered in the clinical workup of the tumor. All microphotographs were captured at 400x magnification. (**D**): DNA abundance, measured as the bias-corrected sequence depth ratio for 10kb bins along the reference genome, appears at distinct levels corresponding to the number of copies per cancer cell. The *RB1*-containing segment displays a low DNA abundance, typical of a deletion. (**E**): The SNP (single nucleotide polymorphism) allele frequency for the *RB1*-containing segment shows distinct allelic imbalance, also consistent with a deletion. The high allele ratio of the pathogenic germline *RB1* variant confirms the retention of the alternative allele in the tumor genome. The estimated average copy number (ploidy) is about 3.6 and the cancer cell fraction is about 60%. Colored dots represent probes located in sarcoma-associated genes.

**Figure 2 cancers-16-03816-f002:**
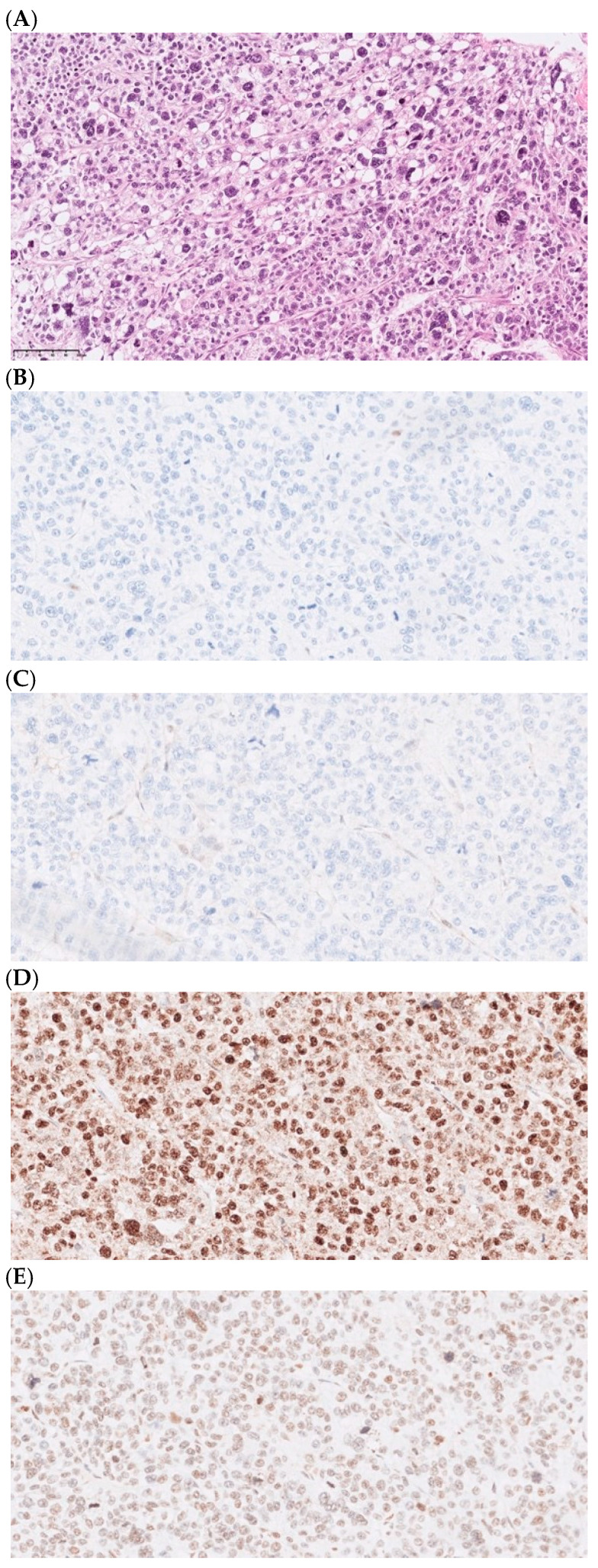
Histopathology and genetics of a Lynch syndrome-associated pleomorphic liposarcoma. Patient 76 was shown to have Lynch syndrome caused by a germline *MLH1* inactivation. Microphotographs of the tumor histology and immunohistochemistry, as performed in the clinical workup. All microphotographs were captured at 400x magnification. (**A**) Routine hematoxylin–eosin stain depicting a pleomorphic liposarcoma and immunohistochemistry with loss of (**B**) MLH1 and (**C**) PMS2 expression, with retained expression of MSH2 (**D**) and MSH6 (**E**), compatible with deficient mismatch repair (dMMR).

**Figure 3 cancers-16-03816-f003:**
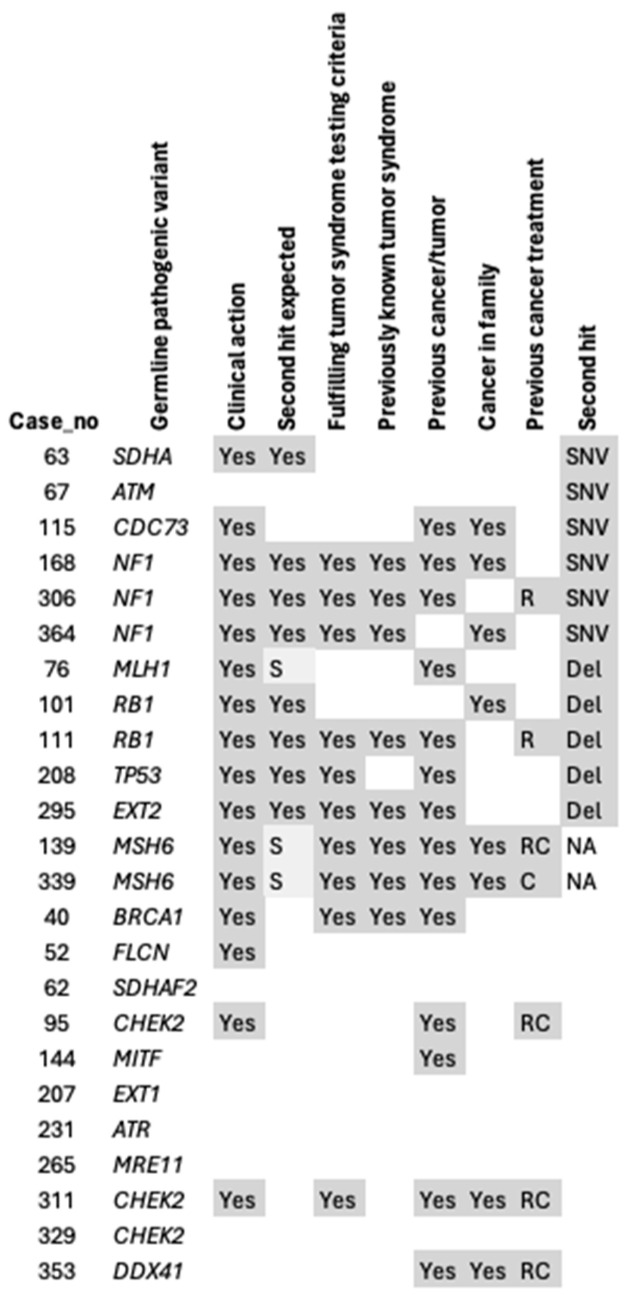
Second hits in relation to phenotype. C: chemotherapy, del: deletion, NA: no tumor tissue available, or second hit detected through immunohistochemistry, R: radiotherapy, RC: radiotherapy combined with chemotherapy, SNV: single nucleotide variant, S: suggested.

**Figure 4 cancers-16-03816-f004:**
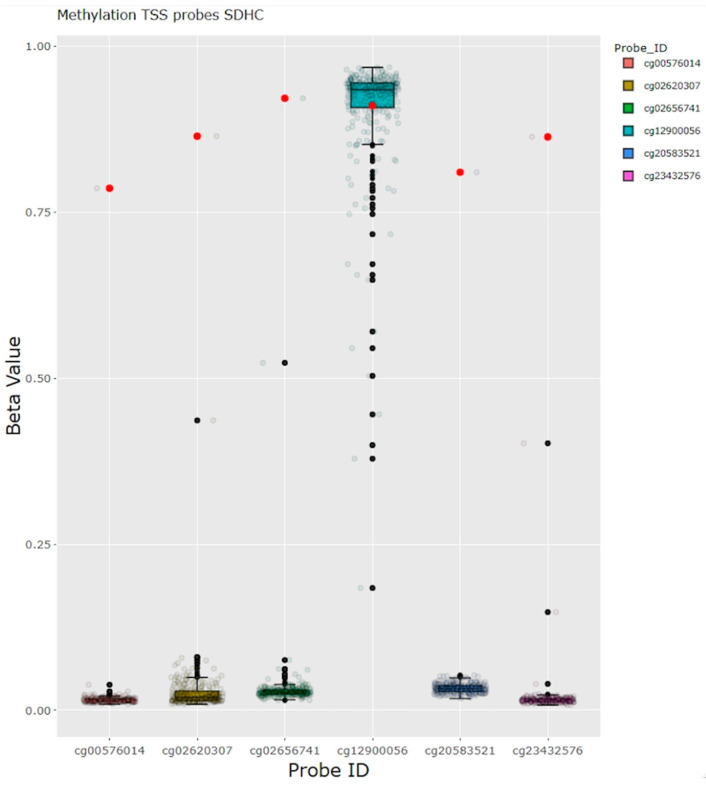
Methylation analysis in a patient with paraganglioma and GIST. Hypermethylation of the *SDHC* promoter in a patient with the clinical diagnosis Carney triad, without any germline findings and no somatic gene dose abnormalities in the *SDHC* region. Red dots represent the beta values of methylation from the case of interest. GIST: gastrointestinal stromal tumor, TSS: transcription start site.

**Table 1 cancers-16-03816-t001:** Genetic findings and clinical data. Diagnoses, germline variant details, clinical data, and second hit detection for the 24 patients with a germline, potentially disease-causing variant.

Case_No	Histological Diagnosis Including Sequencing Results	Germline Finding	Clinical Impact ^A^	Pedigree	Previous Cancer Treatment	Co-Morbidity	Tumor Syndrome Previously Known	Fulfilling Tumor Syndrome Criteria ^B^	Second Hit
40	Leiomyoma	*BRCA1*, NM_007294, c.406del, p.Arg136AspfsTer27	Clinical action	No known cancer	No	OvC 55 y	Yes	Yes	No second hit
52	Osteosarcoma, parosteal	*FLCN*, NM_144606, c.779+1G>T	Clinical action	No known cancer	No	No	No	No	No second hit
62	GIST	*SDHAF2*, NM_017841, c.37-1G>C	Risk factor	No known cancer	No	HT	No	No	No second hit
63	GIST, wild-type	*SDHA*, NM_004168, c.223C>T, p.Arg75Ter	Clinical action	No known cancer	No	DCMP, dementia	No	No	SNV: *SDHA*, NM_004168, c.896G>A, p.Gly299Asp
67	Angiosarcoma	*ATM*, NM_000051, c.7570G>C, p.Ala2524Pro	Risk factor	No known cancer	No	HT	No	No	SNV: *ATM*,NM_000051, c.5188C>T, p.Arg1730Ter
76	Pleomorphic liposarcoma	*MLH1*, NM_000249, c.546-2A>G	Clinical action	No known cancer	No	CRC 52 y, kidney failure, HT	No	No	Deletion *MLH1*
95	Liposarcoma, dedifferentiated	*CHEK2*,NM_001005735, c.1229del, p.Thr410MetfsTer15 ^C^	Clinical action	No known cancer	RT, chemo breast	BrC 65, HT	No	No	No second hit
101	Leiomyosarcoma	*RB1*,NM_000321, c.1981C>T, p.Arg661Trp	Clinical action	Brother’s son suspected RB 4y, mother UtC 50y.	No	No	No	No	Deletion *RB1*
111	Leiomyosarcoma	*RB1*, NM_000321, c.184C>T, p.Gln62Ter	Clinical action	No known cancer	RT OS	RB bilateral 6 months, OS legs multiple during childhood, endometriosis	Yes	Yes	Deletion *RB1*
115	Adenosarcoma, sarcomatous overgrowth	*CDC73*, NM_024529, c.664C>T, p.Arg222Ter	Clinical action	Mother and sister LuC	No	Hypothyreosis, lung embolus	No	No	SNV: *CDC73*, NM_024529, c.25C>T, p.Arg9Ter
139	Radiation-induced sarcoma	*MSH6*,NM_000179, c.2851_2858del, p.Leu951IlefsTer12	Clinical action	Mother OvC, son testis cancer, daughter BC, daughter CxC	RT, chemo breast	BrC 44 y, UtC 48 y	Yes	Yes	No verified second hit
144	GIST	*MITF*, NM_198159, c.1255G>A, p.Glu419Lys	Risk factor	No known cancer	No	UtC 75	No	No	No second hit
168	MPNST	*NF1*, NM_000267, c.1721+3A>C	Clinical action	Sister brain tumor	No	GIST small intestine	Yes	Yes	SNV: *NF1*, NM_000267, c.565A>T, p.Lys189Ter
207	DFSP, fibrosarcoma	*EXT1*, NM_000127, c.1018C>T, p.Arg340Cys	Risk factor	No known cancer	No	No	No	No	No second hit
208	Leiomyosarcoma	*TP53*, NM_001126112, c.503del, p.His168ProfsTer2	Clinical action	No known cancer	Interferone	OS 16 y, LMS 35 y	No	Yes	Deletion *TP53*
231	Liposarcoma (DDLPS)	*ATR*, NM_001184, c.6836dupp.Asn2279LysfsTer4	Risk factor	No known cancer	No	No	No	No	No second hit
265	Leiomyoma	*MRE11*, NM_005591, c.1090C>T, p.Arg364Ter	Risk factor	No known cancer	No	No	No	No	No second hit
295	Secondary peripheral chondrosarcoma	*EXT2*, NM_000401, c.441C>G, p.Tyr147Ter	Clinical action	No known cancer	No	CHS	Yes	Yes	Deletion *EXT2*
306	MPNST	*NF1*, NM_000267, c.4974_4977del, p.Tyr1659ThrfsTer17	Clinical action	No known cancer	RT brain	Intracranial sarcoma 23 y, Scwannoma 24 y, Café-au-laît spots	Yes	Yes	SNV: *NF1*, NM_000267, c.365_371del, p.His122LeufsTer41
311	Synovial chondromatosis	*CHEK2*, NM_001005735, c.1229del, p.Thr410MetfsTer15 ^C^	Clinical action	Sister breast cancer	RT, chemo breast, Tamoxifene	BrC 59 y	No	Yes	No second hit
329	Low-grade parosteal OS	*CHEK2*,NM_001005735, c.1229del, p.Thr410MetfsTer15 ^C^	Risk factor	No known cancer	No	No	No	No	No second hit
339	Soft tissue sarcoma paravertebral ^D^	*MSH6*, NM_000179, c.3261del, p.Phe1088SerfsTer2	Clinical action	Sister OvC, mother BC 59y + UrC 59y + CRC 67y + UtC 53y, maternal grandmother UtC	Chemo	CRC 54 y	Yes	Yes	NA
353	Leiomyosarcoma	*DDX41*, NM_016222, c.415_418dup, p.Asp140GlyfsTer2	Risk factor	Father CRC	RT, chemo kidney	Wilm’s tumor kidney and lung, cholecystectomy, myoma	No	No	No second hit
364	GIST, wild-type	*NF1*, NM_000267, c.6792C>A, p.Tyr2264Ter	Clinical action	Son molecularly verified NF1, no known cancer.	No	Café-au-laît spots	Yes	Yes	SNV: *NF1*, NM_000267, c.3723_3730dup, p.Val1244GlufsTer25

^A^ Clinical decision after considering both genotype and patient phenotype. Clinical action: Known cancer syndrome. The variant was reported to the referring clinician, with recommendations for further genetic counselling. Risk factor: No clinical action, either because national guidelines recommend no surveillance programs or carrier testing in the family given the phenotype of the carrier, or because there is no established cancer syndrome that warrants a surveillance program associated with the variant. ^B^ The patient and his/her family history would fulfill the genetic screening criteria for the syndrome associated with the detected pathogenic germline variant. ^C^ Previously known as NM_007194, c.1100del, p.Thr367MetfsTer15. ^D^ Final classification based on biopsy, since neoadjuvant treatment resulted in a complete pathological response. BC: breast cancer, chemo: chemotherapy, CHS: chondrosarcoma, CRC: colorectal cancer, CxC: cervical cancer, DCMP: dilated cardiomyopathy, HT: hypertension, LMS: leiomyosarcoma, LuC: lung cancer, NA: not applicable, since no tumor tissue, NF1: neurofibromatosis 1, NOS: not otherwise specified, OS: osteosarcoma, OvC: ovarian cancer, RB: retinoblastoma, RT: radiotherapy, SNV: single nucleotide variant, UrC: urothelial cancer, UtC: endometrial cancer (uterus), y: years old (age at diagnosis is specified when known).

**Table 2 cancers-16-03816-t002:** Histopathological diagnoses in the potential cancer syndrome group compared to the total cohort.

	No Germline Finding ^A^		Germline Finding ^B^	
	Number of Cases	Fraction of Total	Number of Cases	Fraction of Total
Soft tissue sarcoma, high-grade	120	42%	11	46%
Soft tissue sarcoma, low-grade	33	11%	1	4%
Soft tissue, benign	40	14%	1	4%
Bone sarcoma, high-grade	13	5%	0	0%
Bone sarcoma, low-grade	7	2%	3	13%
Bone, benign	5	2%	0	0%
GIST	36	13%	4	17%
Gynaecological tract	34	12%	4	17%
Sum	288		24	

^A^ Subcohort with no pathogenic variants in a gene associated with a hereditary cancer syndrome (N = 288). ^B^ Subcohort with a detected pathogenic variant in a gene associated with a hereditary cancer syndrome (N = 24). GIST: gastrointestinal stromal tumor.

## Data Availability

Data generated in this study are available from the corresponding author upon request. Raw data is not publicly available, because it contains information that could compromise research participants’ anonymity.

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
