# Peer review of "Prospective Screening of Cancer Syndromes in Patients with Mesenchymal Tumors"

_cancers, 2024, doi:10.3390/cancers16223816_

Round 1

Reviewer 1 Report

Comments and Suggestions for Authors

General comment

 I read with great attention the manuscript entitled "Prospective Screening of Cancer Syndromes in Patients with Mesenchymal Tumors” by Ingegerd Öfverholm et al

The authors have highlighted the utility of second hit analyses in standard clinical practice for identifying syndrome-related tumors, emphasizing the importance of distinguishing between relevant and unrelated germline mutations.

Overall, the results are compelling.

Additionally, the following observations are made to provide a better understanding of the study for readers of this journal.

- the SUPPLEMENTARY TABLE 1: Genes included in the in-silico germline list. Please arrange the list of genes in the table in alphabetical order.

-   The abbreviation 'gastrointestinal stromal tumor' (GIST) is fully defined only in the Discussion section; it should be defined the first time it is used. Additionally, in Supplementary Table 2, the meaning for 'GIST' and 'G' are missing below the table.

-   Table 1, 'Genetic Findings and Clinical Data,' should be better formatted for clarity. As it currently stands, it is hard to read. Also, in some cases there is information on the age of the patient and for others no, should also be uniformized the use of “y” to indicate years old. In the case 339 where it is “Soft tissue sarcoma paravertebral*** should be “Soft tissue sarcoma paravertebral****

-   The information on lines 217 to 221 needs to be re-written for clarification.

-   “…MMR (mismatch repair)” should be “mismatch repair (MMR)”

-   In Figure 1, each panel could be labeled with a letter for identification, which would make it easier to follow the legend. The same for Figure 2.

-   The authors should follow the instructions for authors regarding “Acronyms/Abbreviations/Initialisms should be defined the first time they appear in each of three sections: the abstract; the main text; the first figure or table. When defined for the first time, the acronym/abbreviation/initialism should be added in parentheses after the written-out form.”

-   In the legend of Figure 3 the meaning for RC is missing.

-   The title for the Supplementary Table 4 is a little confusing: “Second hits for all pathogenic and likely pathogenic variants not causative for a tumor syndrome in the patient, but for a non-tumor syndrome”

-   The phrase “ATM is DNA-repair gene, classified as a moderately-penetrant breast cancer gene.” Should be changed for clarification and a reference should be added.

-   A reference is needed after the information “The detection of biallelic loss of ATM in an angiosarcoma tumor contributes to this growing knowledge of ATM in tumorigenesis”

-   In the Discussion the authors mention that “The ACMG criteria (17), including statistical correlation between diagnosis frequency and carrier frequency, bioinformatic information such as how conserved the affected amino acid is, etcetera, are widely used in the clinic.” Consider the use of recommendations by the American College of Medical Genetics and Genomics and the Association for Molecular Pathology (ACMG/AMP). 

Author Response

ANSWERS TO REVIEWER 1 

  1. SUPPLEMENTARY TABLE 1: Genes included in the in-silico germline list. Please arrange the list of genes in the table in alphabetical order. 

Thank you for pointing this out. It has been updated as suggested, with all gene names in alphabetical order.  

  1. The abbreviation 'gastrointestinal stromal tumor' (GIST) is fully defined only in the Discussion section; it should be defined the first time it is used. Additionally, in Supplementary Table 2, the meaning for 'GIST' and 'G' are missing below the table.  

We agree with the comment, and we have corrected this by fully identifying the abbreviation on line 174 in Results, added the description in the legend for Table 2, and only used the abbreviation on line 451. Also, we have changed the “G” for GIST (column “Malignant sarcoma?”) in Supplementary Table 2.  

  1. Table 1, 'Genetic Findings and Clinical Data,' should be better formatted for clarity. As it currently stands, it is hard to read. Also, in some cases there is information on the age of the patient and for others no, should also be uniformized the use of “y” to indicate years old. In the case 339 where it is “Soft tissue sarcoma paravertebral*** should be “Soft tissue sarcoma paravertebral**** 

Yes, we agree with all corrections suggested. We have decreased the font size in the tables, added “y” to patient 95 and 144 in Table 1, replaced the asterisks with letters, and corrected the “soft tissue sarcoma paravertebral”. Please see the revised Table 1 below.  

Regarding the cancer diagnoses with no age specified, the patients had no knowledge about the exact age, and it was not specified in available charts. We added a clarification in the figure legend for Table 1: “(age of diagnosis specified when known)”. 

Table 1:

Case_no 

Histological diagnosis including sequencing results 

Germline finding 

Clinical impactA 

Pedigree 

Previous cancer treatment 

Co-morbidity 

Tumor syndrome previously known 

Fulfilling tumor syndrome criteriaB 

Second hit 

40 

Leiomyoma 

BRCA1,NM_007294,c.406del,p.Arg136AspfsTer27 

Clinical action 

No known cancer 

No 

OvC 55y 

Yes 

Yes 

No second hit 

52 

Osteosarcoma, paraosteal 

FLCN,NM_144606,c.779+1G>T 

Clinical action 

No known cancer 

No 

No 

No 

No 

No second hit 

62 

GIST 

SDHAF2,NM_017841,c.37-1G>C 

Risk factor 

No known cancer 

No 

HT 

No 

No 

No second hit 

63 

GIST, wild-type 

SDHA,NM_004168,c.223C>T,p.Arg75Ter 

Clinical action 

No known cancer 

No 

DCMP, dementia 

No 

No 

SNV: SDHA,NM_004168,c.896G>A,p.Gly299Asp 

67 

Angiosarcoma 

ATM,NM_000051,c.7570G>C,p.Ala2524Pro 

Risk factor 

No known cancer 

No 

HT 

No 

No 

SNV: ATM,NM_000051,c.5188C>T,p.Arg1730Ter 

76 

Pleomorphic liposarcoma 

MLH1,NM_000249,c.546-2A>G 

Clinical action 

No known cancer 

No 

CRC 52y, kidney failure, HT 

No 

No 

Deletion MLH1 

95 

Liposarcoma, dedifferentiated 

CHEK2,NM_001005735,c.1229del,p.Thr410MetfsTer15C 

Clinical action 

No known cancer 

RT, chemo breast 

BrC 65, HT 

No 

No 

No second hit 

101 

Leiomyosarcoma 

RB1,NM_000321,c.1981C>T,p.Arg661Trp 

Clinical action 

Brother's son suspected RB 4y, mother UtC 50y.  

No 

No 

No 

No 

Deletion RB1 

111 

Leiomyosarcoma 

RB1,NM_000321,c.184C>T,p.Gln62Ter 

Clinical action 

No known cancer 

RT OS 

RB bilateral 6 months, OS legs multiple during childhood, endometriosis 

Yes 

Yes 

Deletion RB1 

115 

Adenosarcoma, sarcomatous overgrowth 

CDC73,NM_024529,c.664C>T,p.Arg222Ter 

Clinical action 

Mother and sister LuC 

No 

Hypothyreosis, lung emobolus 

No 

No 

SNV: CDC73,NM_024529,c.25C>T,p.Arg9Ter 

139 

Radiation induced sarcoma 

MSH6,NM_000179,c.2851_2858del,p.Leu951IlefsTer12 

Clinical action 

Mother OvC, son testis cancer, daughter BC, daughter CxC 

RT, chemo breast 

BrC 44y, UtC 48y 

Yes 

Yes 

No verified second hit 

144 

GIST 

MITF,NM_198159,c.1255G>A,p.Glu419Lys 

Risk factor 

No known cancer 

No 

UtC 75 

No 

No 

No second hit 

168 

MPNST 

NF1,NM_000267,c.1721+3A>C 

Clinical action 

Sister brain tumor 

No 

GIST small intestine 

Yes 

Yes 

SNV: NF1,NM_000267,c.565A>T,p.Lys189Ter 

207 

DFSP, fibrosarcoma 

EXT1,NM_000127,c.1018C>T,p.Arg340Cys 

Risk factor 

No known cancer 

No 

No 

No 

No 

No second hit 

208 

Leiomyosarcoma 

TP53,NM_001126112,c.503del,p.His168ProfsTer2 

Clinical action 

No known cancer 

Interferone 

OS 16y, LMS 35y 

No 

Yes 

Deletion TP53 

231 

Liposarcoma (DDLPS) 

ATR,NM_001184,c.6836dupp.Asn2279LysfsTer4 

Risk factor 

No known cancer 

No 

No 

No 

No 

No second hit 

265 

Leiomyoma 

MRE11,NM_005591,c.1090C>T,p.Arg364Ter 

Risk factor 

No known cancer 

No 

No 

No 

No 

No second hit 

295 

Secondary peripheral chondrosarcoma 

EXT2,NM_000401,c.441C>G,p.Tyr147Ter 

Clinical action 

No known cancer 

No 

CHS 

Yes 

Yes 

Deletion EXT2 

306 

MPNST 

NF1,NM_000267,c.4974_4977del,p.Tyr1659ThrfsTer17 

Clinical action 

No known cancer 

RT brain 

Intracranial sarcoma 23y, Scwannoma 24y, Café-au-laît spots 

Yes 

Yes 

SNV: NF1,NM_000267,c.365_371del,p.His122LeufsTer41 

311 

Synovial chondromatosis 

CHEK2,NM_001005735,c.1229del,p.Thr410MetfsTer15C

Clinical action 

Sister breast cancer 

RT, chemo breast, Tamoxifene 

BrC 59y 

No 

Yes 

No second hit 

329 

Low-grade paraosteal OS 

CHEK2,NM_001005735,c.1229del,p.Thr410MetfsTer15C 

Risk factor 

No known cancer 

No 

No 

No 

No 

No second hit 

339 

Soft tissue sarcoma paravertebralD 

MSH6,NM_000179,c.3261del,p.Phe1088SerfsTer2 

Clinical action 

Sister OvC, mother BC 59y + UrC 59y + CRC 67y + UtC 53y, maternal grandmother UtC 

Chemo 

CRC 54y 

Yes 

Yes 

NA 

353 

Leiomyosarcoma 

DDX41,NM_016222,c.415_418dup,p.Asp140GlyfsTer2 

Risk factor 

Father CRC 

RT, chemo kidney 

Wilm's tumor kidney and lung, cholecystectomy, myoma 

No 

No 

No second hit 

364 

GIST, wild-type 

NF1,NM_000267,c.6792C>A,p.Tyr2264Ter 

Clinical action 

Son molecularly verified NF1, no known cancer. 

No 

Café-au-laît spots 

Yes 

Yes 

SNV: NF1,NM_000267,c.3723_3730dup,p.Val1244GlufsTer25 

  1. The information on lines 217 to 221 needs to be re-written for clarification. 

Thank you for this comment. We have re-written the paragraph (lines 222-229 in the current submission):  

“The remaining 8 pathogenic and potentially actionable germline variants were not reported to the treating clinicians, in accordance with ACMG/AMP criteria and Swedish National Guidelines. These 8 variants were either associated to potential hereditary tumor syndromes without any surveillance recommendations based on patient phenotype and pedigree (variants in the genes CHEK2, EXT1, ATM), or they were weakly associated to a condition potentially increasing the risk for cancer without being likely causative for a cancer syndrome in the specific case (variants in the genes MRE11, BRIP1, ATR, DDX41, MITF, SDHAF2).” 

  1. …MMR (mismatch repair)” should be “mismatch repair (MMR)” 

This has been corrected on lines 237-238. 

  1. In Figure 1, each panel could be labeled with a letter for identification, which would make it easier to follow the legend. The same for Figure 2. 

Figure 1 and Figure 2 have been updated as suggested, and we have added technical information in the figure legends:

“Figure 1. Histopathology and genetics of a RB1-germline related leiomyosarcoma. Patient 101, carrying a germline RB1 pathogenic variant causing hereditary retinoblastoma syndrome. The tumor harbored a deletion of the RB1 locus (13q), resulting in loss of heterozygosity and biallelic inactivation. A-C: Microphotographs shows (A) routine hematoxylin-eosin stain of a leiomyomatous tumor with high grade atypia and immunohistochemistry showing (B) positivity for desmin and (C) loss of Rb immunoreactivity (single cells with retained Rb expression are tumor infiltrating immune cells) as performed in the clinical workup of the tumor. All microphotographs were captured at 400X magnification. D: DNA abundance measured as bias-corrected sequence depth ratio for 10kb bins along the reference genome, appear at distinct levels corresponding to a number of copies per cancer cell. The RB1 containing segment displays a low DNA abundance typical for deletion. E: Single nucleotide polymorphism (SNP) allele frequency for the RB1 containing segment shows distinct allelic imbalance, also consistent with a deletion. The high allele ratio of the pathogenic germline RB1 variant confirms retention of the alternative allele in the tumor genome. The estimated average copy number (ploidy) is about 3.6 and the cancer cell fraction is about 60%.”

 “Figure 2. Histopathology and genetics of a Lynch syndrome associated pleomorphic liposarcoma. Patient 76 was shown to have Lynch syndrome caused by a germline MLH1 inactivation. Microphotographs of the tumor histology and immunohistochemistry as performed in the clinical workup. All microphotographs were captured at 400X magnification. (A) Routine hematoxylin-eosin stain depicting a pleomorphic liposarcoma and immunohistochemistry with loss of (B) MLH1 and (C) PMS2 expression, with retained expression of MSH2 (D) and MSH6 (E), compatible with deficient mismatch repair (dMMR).” 

  1. The authors should follow the instructions for authors regarding “Acronyms/Abbreviations/Initialisms should be defined the first time they appear in each of three sections: the abstract; the main text; the first figure or table. When defined for the first time, the acronym/abbreviation/initialism should be added in parentheses after the written-out form. 

Care has been taken to follow these instructions, and the reviewer has kindly pointed out that it needed to be corrected for GIST, MMR, RC, and ACMG throughout this review. Also, we have made corrections for SNP.

  1. In the legend of Figure 3 the meaning for RC is missing. 

This has been added in the figure legend:  
“RC: radiotherapy combined with chemotherapy” 

  1. The title for the Supplementary Table 4 is a little confusing: “Second hits for all pathogenic and likely pathogenic variants not causative for a tumor syndrome in the patient, but for a non-tumor syndrome” 

We agree, and the title has therefore been changed into “Second hits for all pathogenic and likely pathogenic non-tumor syndrome variants”.

  1. The phrase “ATM is DNA-repair gene, classified as a moderately-penetrant breast cancer gene.” Should be changed for clarification and a reference should be added. 

We have clarified the sentence on lines 376-378 to:  

“ATM is a tumor suppressor gene with multiple protein functions, such as DNA-repair and cell cycle regulation. In current clinical practice, this gene is classified as a moderate penetrance gene associated with increased risk for breast cancer [31].” 

  1. A reference is needed after the information “The detection of biallelic loss of ATM in an angiosarcoma tumor contributes to this growing knowledge of ATM in tumorigenesis” 

The sentence was referring to the findings in the present study, and we have made this clearer by changing the sentence into:  

“The detection of biallelic loss of ATM in an angiosarcoma tumor in our study contributes to the growing knowledge about ATM in tumorigenesis.” 

on lines 388-390.  

  1. In the Discussion the authors mention that “The ACMG criteria (17), including statistical correlation between diagnosis frequency and carrier frequency, bioinformatic information such as how conserved the affected amino acid is, etcetera, are widely used in the clinic.” Consider the use of recommendations by the American College of Medical Genetics and Genomics and the Association for Molecular Pathology (ACMG/AMP).  

This has been corrected at first point of appearance in the main text, in the Methods section on lines 115-116:  

“… phenotype and the American College of Medical Genetics and Genomics and Association for Molecular Pathology (ACMG/AMP) criteria…” 

and all mentions of “ACMG” have been updated to “ACMG/AMP” in the rest of the text.  

Reviewer 2 Report

Comments and Suggestions for Authors

Comments and Suggestions for the authors of Cancers-3276678:

The study investigated the genetic predisposition to tumor syndromes in patients with mesenchymal tumors, which are rare and have limited established causes. The authors utilized a comprehensive methodology that integrates prospective germline analysis with tumor whole genome sequencing, as well as transcriptome and methylome analyses, to identify cancer syndromes. Their findings reveal both known and novel mutations associated with an increased risk of tumors, thereby advancing the field of genetic oncology and potentially impacting clinical practice, particularly in targeted screening and treatment. This research addresses a critical gap in understanding the genetic predisposition to mesenchymal tumors. Furthermore, insights gained from second-hit analyses in standard care may significantly change clinical management practices for patients with these tumors.

Suggestions:

·         In Figures 1 and 2, please use a standardized labeling method such as A, B, C… for the panels, and provide more detailed descriptions of each experiment.

·         Strengthen the validation of novel mutations by conducting additional in vitro or in vivo studies, or by incorporating existing datasets to confirm the functional impact of the identified mutations. For example, an analysis of large-scale genetic databases, such as the Catalogue of Somatic Mutations in Cancer (COSMIC) or The Cancer Genome Atlas (TCGA), to determine if the identified variants are reported in other cancer types.

·         Include longitudinal follow-up data, if feasible, to provide information on the long-term clinical outcomes of patients with identified germline variants, enhancing the clinical relevance of the findings and offering further support for the results.

·         Clinical Correlations: Establishing correlations between the presence of these variants and clinical outcomes in patients with different cancer types, which could provide real-world evidence of their role in cancer susceptibility. Such as, what are the implications of these mutations in the mesenchymal subtype of glioblastoma?

Author Response

ANSWERS TO REVIEWER 2 

  1. In Figures 1 and 2, please use a standardized labeling method such as A, B, C… for the panels, and provide more detailed descriptions of each experiment. 

We have, as suggested, changed to alphabetical labelling. Also, we have updated the figure legend for Figure 1:

“Figure 1. Histopathology and genetics of a RB1-germline related leiomyosarcoma. Patient 101, carrying a germline RB1 pathogenic variant causing hereditary retinoblastoma syndrome. The tumor harbored a deletion of the RB1 locus (13q), resulting in loss of heterozygosity and biallelic inactivation. A-C: Microphotographs shows (A) routine hematoxylin-eosin stain of a leiomyomatous tumor with high grade atypia and immunohistochemistry showing (B) positivity for desmin and (C) loss of Rb immunoreactivity (single cells with retained Rb expression are tumor infiltrating immune cells) as performed in the clinical workup of the tumor. All microphotographs were captured at 400X magnification. D: DNA abundance measured as bias-corrected sequence depth ratio for 10kb bins along the reference genome, appear at distinct levels corresponding to a number of copies per cancer cell. The RB1 containing segment displays a low DNA abundance typical for deletion. E: Single nucleotide polymorphism (SNP) allele frequency for the RB1 containing segment shows distinct allelic imbalance, also consistent with a deletion. The high allele ratio of the pathogenic germline RB1 variant confirms retention of the alternative allele in the tumor genome. The estimated average copy number (ploidy) is about 3.6 and the cancer cell fraction is about 60%.”

And Figure 2:
“Figure 2. Histopathology and genetics of a Lynch syndrome associated pleomorphic liposarcoma. Patient 76 was shown to have Lynch syndrome caused by a germline MLH1 inactivation. Microphotographs of the tumor histology and immunohistochemistry as performed in the clinical workup. All microphotographs were captured at 400X magnification. (A) Routine hematoxylin-eosin stain depicting a pleomorphic liposarcoma and immunohistochemistry with loss of (B) MLH1 and (C) PMS2 expression, with retained expression of MSH2 (D) and MSH6 (E), compatible with deficient mismatch repair (dMMR).” 

  1. Strengthen the validation of novel mutations by conducting additional in vitro or in vivo studies, or by incorporating existing datasets to confirm the functional impact of the identified mutations. For example, an analysis of large-scale genetic databases, such as the Catalogue of Somatic Mutations in Cancer (COSMIC) or The Cancer Genome Atlas (TCGA), to determine if the identified variants are reported in other cancer types. 

Thank you for bringing this up. The novel variants we report all have well-established deleterious effects and are reported as pathogenic by multiple submitters in ClinVar, although not in association with inherited sarcoma cancer syndromes. To further strengthen the assumption of a biallelic deleterious effect in the tumors for these novel variants, we have conducted a database search for the somatic second hit variants from our study in other cancer cohorts, as suggested by the reviewer, and relevant results are summarized in the discussion. We have also investigated the prevalence of lesions in these genes in soft tissue cohorts. This does indeed add some more insights about functional consequences of the variants. 

We have added this paragraph in the Methods section, on lines 141-145:  

“Germline variants with no previously known association to the phenotype but detectable same-gene second hits were considered novel, and these genes were further checked in the Catalogue Of Somatic Mutations In Cancer (COSMIC) database and The Cancer Genome Atlas (TCGA) datasets for further assessment of potential impact in soft tissue tumors [20] [21].”. 

For each gene of interest, we have added a short summary of the database search results:  

For the ATM gene, we have added this information on lines 382-388: 

“The somatic second hit detected in our patient, c.5188C>T, is a well-established truncating pathogenic germline variant [16]. In the COSMIC database, somatic ATM variants are reported in 8% (82/1077) of soft tissue tumors, of which 16 are angiosarcomas. The c.5188C>T variant is reported in 13 cases (carcinomas (N=9), malignant melanomas (N=3), and one carcinoid) [20]. Aberrations in ATM are present in 5% (12/255) of the PanCancer Atlas Sarcoma dataset, mostly in myxofibrosarcomas [21].”  

For the CDC73 gene, we have added this information on lines 424-427:  

“While the truncating somatic variant detected in our patient has been reported solely in parathyroid carcinoma (N=4) in the COSMIC database, other somatic variants in CDC73 are reported in 2% (30/1679) of soft tissue tumors [20].” 

For the KCNQ1 gene, we have added this information on lines 402-406:  

“Both the germline and somatic variants detected in our patient are known pathogenic variants associated to long QT syndrome, and none of these specific variants are reported in tumor samples in the COSMIC database. However, other KCNQ1 variants are reported in a variety of tumor types, including 4% (33/867) of soft tissue tumors, mainly in unclassified sarcoma from fibrous tissue/uncertain origin (23/33) [20].” 

For the POLG gene, we have added this information on lines 394-399: 

“The POLG germline variant identified in our study is reported as pathogenic in mitochondrial disease cohorts by multiple submitters in ClinVar [16], and somatic variants in POLG are reported in 1% (10/902) of soft tissue tumors in COMIC, of which and 30% (3/10) are GISTs [20]. A second hit in GIST is intriguing, and further studies are needed to determine the functional role of biallelic POLG inactivation in tumor development.” 

  1. Include longitudinal follow-up data, if feasible, to provide information on the long-term clinical outcomes of patients with identified germline variants, enhancing the clinical relevance of the findings and offering further support for the results. 

We agree with the reviewer, longitudinal follow-up data would add significant value to the findings. However, in this prospective study, all cases have a follow-up time shorter than 5 years (median 2 years), which would be too short to draw any conclusions on differences in outcomes between groups. A future project will be to describe clinical outcome (alive without disease, alive with disease, deceased) for these patients.  

  1. Clinical Correlations: Establishing correlations between the presence of these variants and clinical outcomes in patients with different cancer types, which could provide real-world evidence of their role in cancer susceptibility. Such as, what are the implications of these mutations in the mesenchymal subtype of glioblastoma? 

This is an interesting comment indeed. We have checked the cBioPortal set of non-redundant studies, to get a picture of clinical outcome when any variant in the genes of interest is detected, since there is not enough data on the specific variants to make any reliable survival predictions. In the curated set of non-redundant studies displayed in cBioPortal, the overall survival times for patients with either ATM, POLG, KCNQ1, MLH1 or MSH6 somatic variants versus no variants in the same gene are not significantly altered. Variants in CDC73 are associated to a longer overall survival time. While these mutations (in a somatic setting) lacked prognostic value in a pan-cancer analysis, we could not exclude it either. We did not include these additional analyses given i) the risk of type-2 statistical errors and ii) our inability to adjust for other key factors (i.e. clinical stage) when doing these comparisons. While this question is very important, a larger effort of data analysis would be required to truly answer this hypothesis, which we believe lies outside the scope of this paper. 

Round 2

Reviewer 2 Report

Comments and Suggestions for Authors

Thank you for your attention to my suggestion. I agree to publish this version of the manuscript.